# The OsteoSense Imaging Agent Identifies Organ-Specific Patterns of Soft Tissue Calcification in an Adenine-Induced Chronic Kidney Disease Mouse Model

**DOI:** 10.3390/ijms26178525

**Published:** 2025-09-02

**Authors:** Gréta Lente, Andrea Tóth, Enikő Balogh, Dávid Máté Csiki, Béla Nagy, Árpád Szöőr, Viktória Jeney

**Affiliations:** 1MTA-DE Lendület Vascular Pathophysiology Research Group, Research Centre for Molecular Medicine, Faculty of Medicine, University of Debrecen, 4032 Debrecen, Hungary; lente.greta@med.unideb.hu (G.L.); andrea.toth@med.unideb.hu (A.T.); balogh.eniko@med.unideb.hu (E.B.); csiki.david.mate@gmail.com (D.M.C.); 2Molecular Cell and Immunobiology Doctoral School, University of Debrecen, 4032 Debrecen, Hungary; 3Department of Laboratory Medicine, Faculty of Medicine, University of Debrecen, 4032 Debrecen, Hungary; bela.nagy@med.unideb.hu; 4Department of Biophysics and Cell Biology, Faculty of Medicine, University of Debrecen, 4032 Debrecen, Hungary; akuka@med.unideb.hu

**Keywords:** OsteoSense, chronic kidney disease, soft tissue calcification, vascular calcification, ectopic mineralization

## Abstract

Extra-osseous calcification refers to the pathological deposition of calcium salts in soft tissues. Its most recognized forms affect the cardiovascular system, leading to vascular and heart valve calcifications. This process is active and regulated, involving the phenotype transition of resident cells into osteo/chondrogenic lineage. Chronic kidney disease (CKD) patients frequently suffer from vascular and other soft tissue calcification. OsteoSense dyes are fluorescent imaging agents developed to visualize calcium deposits during bone formation. In addition to its application in bone physiology, it has been used to detect vascular smooth muscle cell calcification in vitro and to evaluate calcification ex vivo. Here, we investigated CKD-associated soft tissue calcification by applying OsteoSense in vivo. CKD was induced by a diet containing adenine and elevated phosphate. OsteoSense (80 nmol/kg body weight) was injected intravenously through the retro-orbital venous sinus 18 h before the measurement on an IVIS Spectrum In Vivo Imaging System. OsteoSense staining detected calcium deposition in the aorta, kidney, heart, lung, and liver in CKD mice. On the other hand, no calcification occurred in the brain, eye, or spleen. OsteoSense positivity in the calcified soft tissues in CKD mice was associated with increased mRNA levels of osteo/chondrogenic transcription factors. Our findings demonstrate that OsteoSense is a sensitive and effective tool for detecting soft tissue calcification in vivo, and may be particularly valuable for studies of CKD-related ectopic calcification.

## 1. Introduction

Chronic kidney disease (CKD) is a progressive condition that affects more than 10% of the general population [1]. Patients with CKD have an increased risk of cardiovascular diseases (CVD) due to premature aging of the vascular system, heart, as well as the calcification of soft tissues. As a result, CKD is strongly associated with high cardiovascular mortality and morbidity [2].

Vascular calcification (VC) is characterized by the abnormal accumulation of calcium salts in the walls of blood vessels. VC shares many similarities with the physiological process of bone formation [3]. VC is a cell-mediated process actively regulated by various pathological stimuli, including aging, hypertension, diabetes mellitus, and chronic kidney disease (CKD) [4]. As kidney function declines, serum levels of phosphate and calcium increase. Additionally, CKD is associated with the loss of calcification inhibitors. The imbalance between calcification inducers and inhibitors triggers a phenotypic switch in vascular smooth muscle cells (VSMCs), transforming them into osteoblast-like cells [5]. During the osteogenic phenotype switch, VSMCs lose their contractile markers and begin to express osteoblast-specific transcriptional regulators, including runt-related transcription factor 2 (Runx2), SRY-box transcription factor 9 (Sox9), Msh homeobox 2 (Msx2), and bone morphogenetic protein 2 (BMP2) [6,7].

There are two classical, widely used staining methods for detecting soft tissue calcification: von Kossa staining (VKS) and alizarin red S staining (ARS). VKS was developed by the Hungarian physician Julius von Kossa [8]. VKS is based on the substitution of calcium ions with silver ions, followed by the reduction of silver ions and the precipitation of metallic silver, producing a brownish or black coloration at the site of calcification. VKS is an inexpensive and simple method widely used to identify abnormal calcium deposition in tissues and calcifying cultured cells. However, the technique has limitations, including poor specificity and black deposits covering the structural details of the stained tissue [8].

Another widely used staining method for detecting calcification is ARS, which was developed in 1869 by two German chemists, Graebe and Liebermann, to detect calcification through binding to calcium. Similar to VKS, ARS detects macrocalcification but lacks sensitivity for microcalcification [9].

OsteoSense dyes (680EX, 750EX, 800EX) are fluorescently labeled bisphosphonate compounds used to detect and image calcium deposits in various biological settings. First, Zilberman et al. applied OsteoSense to investigate bone remodeling in mice, providing a novel non-invasive method for imaging bone formation [10]. Then, Moester et al. showed that OsteoSense is useful in detecting and quantifying in vitro matrix mineralization of osteoblasts [11]. Beyond its use in the skeletal system, OsteoSense 680EX has also been employed to detect urinary tract stones, particularly calcium oxalate–struvite, calcium phosphate–struvite, and ammonium urate–calcium oxalate–calcium phosphate stones [12]. Additionally, OsteoSense senses joint ankylosis in ank/ank mice and microcalcifications associated with breast cancer [13,14]. In connection with vascular calcification, it has been shown that OsteoSense can detect calcifying VSMCs in vitro [15,16] and aorta calcifications on aortic sections derived from a mouse model of Marfan syndrome [17]. OsteoSense staining has been used by our group and others to visualize aortic calcification in vivo in mouse models using whole tissues [18,19].

All the previous studies suggest that OsteoSense is a highly sensitive tool for detecting calcification across various tissues and cell types. This study aimed to evaluate the utility of the OsteoSense imaging agent for detecting and quantifying soft tissue calcification in a mouse model of CKD. Additionally, we investigated whether OsteoSense signal correlates with markers of active osteogenic differentiation in the calcifying soft tissues. 

## 2. Results

### 2.1. Intravenous Injection of OsteoSense 750EX Detects Calcification in the Aorta and Kidney in Adenine-Induced CKD Mice

To investigate CKD-associated soft tissue calcification, we induced CKD in male C57BL/6 mice by applying a two-phase diet containing adenine and increased phosphate as described previously by Tani et al. [20]. We randomly assigned the mice to control (CTRL) and CKD groups (n = 12/group). The experimental setting is shown in Figure 1a. Control mice were fed a standard chow containing 0.3% phosphate for 9 weeks. CKD mice were fed CKD diet #1 containing adenine (0.2%) and moderately elevated phosphate (0.7%) for 6 weeks, followed by a CKD diet #2 containing adenine (0.2%) and high phosphate (1.8%) for 3 weeks (Figure 1a). After the nine experimental weeks, the mice were anesthetized with isoflurane and injected with OsteoSense dye (80 nmol/kg body weight) intravenously through the retro-orbital venous sinus. Eighteen hours later, the mice were killed with CO_2_ inhalation. We took blood by cardiac puncture, perfused the mice with 5 mL of ice-cold PBS, and collected the organs for IVIS analysis (n = 8) and RNA isolation (n = 4) (Figure 1b).

Serum urea and creatinine levels were elevated in CKD samples compared to controls, indicating a decline in kidney function in adenine-treated mice (Figure 1c,d). Serum phosphate levels were also elevated in CKD samples compared to controls (Figure 1e), supposedly due to the high-phosphate diet and the decreased ability of the kidney to excrete excess phosphate. In contrast, serum calcium levels remained unchanged (Figure 1f).

Immediately after harvesting, the aortas were thoroughly cleaned of fat and connective tissue and analyzed ex vivo using an IVIS Imager within 4 h of collection. OsteoSense signal intensity was significantly higher in the aortas of CKD mice compared to CTRLs. (Figure 2a,b). OsteoSense staining revealed extensive calcification along the entire aorta in CKD mice, with the highest fluorescent intensity observed in the abdominal region, specifically at the site of the renal artery origin and the arch (Figure 2a).

Then, we performed H&E, VKS, and ARS staining on aorta cross-sections of CTRL and CKD mice (Figure 2c). We did not detect any structural changes or calcification in the aorta of CKD mice with H&E staining and VKS, respectively (Figure 2c). However, we detected a few calcification nodules with ARS (Figure 2c).

Next, we evaluated kidney calcification with OsteoSense staining. We detected intensive OsteoSense staining in the kidneys of CKD mice but not in the CTRLs (Figure 3a,b). OsteoSense positivity in CKD kidneys was associated with structural alterations, including atrophy and fibrosis, typical morphological manifestations of CKD (Figure 3a). H&E staining revealed tubular dilatation in the kidneys of CKD mice compared to CTRLs (Figure 3c). We detected kidney calcification with both VKS and ARS in CKD mice but not in the CTRLs (Figure 3c). Additionally, in a pilot experiment, we investigated whether OsteoSense can detect calcification in paraffin-embedded kidney samples derived from CKD mice. We found that OsteoSense positivity coincided completely with the traditional VKS and AR staining (unpublished results).

### 2.2. CKD Is Associated with the Elevation of Osteogenic Marker mRNA Levels in the Aorta and Kidney

Then, we determined the mRNA levels of four transcription factors, Runx2, Sox9, Msx2, and Bmp2, which play regulatory roles in the osteogenic transition of VSMCs. We detected a mild increase in Runx2 and strong upregulation of Msx2 and Bmp2 mRNA levels in the aorta samples derived from CKD mice compared to aortas from CTRLs (Figure 4a). All four osteogenic marker mRNAs were upregulated in the kidney obtained from the CKD mice; among them, Sox9 and Runx2 showed the highest increases compared to kidneys from CTRLs (Figure 4b).

### 2.3. CKD Triggers Calcification and an Increase in Osteogenic Marker mRNA Levels in the Heart, Lung, and Liver

Next, we examined whether CKD is associated with osteogenic processes in other vital organs, such as the heart, lungs, and liver. Within four hours of harvesting, we analyzed the OsteoSense fluorescence intensity of these organs. We found intensive OsteoSense staining in the hearts and lungs of CKD mice but not in the CTRLs (Figure 5a–d). A lower-intensity fluorescent signal originated from the entire heart and lung parenchyma (Figure 5a,c). OsteoSense staining intensity was the highest at the regions where vessels enter and exit the heart and the lungs (Figure 5a,c). Compared to CTRLs, liver from CKD mice showed significantly higher OsteoSense staining (Figure 5e,f). However, the fluorescence staining intensity in the CKD liver was about two magnitudes lower than in the heart or lungs (Figure 5). Calcification was located centrally in the liver, with the peripheral regions of the lobes showing no staining (Figure 5e). Moreover, CKD was associated with increased mRNA expression of Runx2 and Bmp2 in the heart and liver, as well as elevation of Sox9 mRNA level in the lungs (Figure 6a–c). Interestingly, although Msx2 was strongly upregulated in calcifying aorta, it was undetectable in the liver, heart, and lungs.

### 2.4. CKD Does Not Induce Significant Calcification in the Brain, Eye, and Spleen

We also investigated calcification in the brains, eyes, and spleens of CTRL and CKD mice. We detected spotty microcalcifications in the brains of CTRL and CKD mice, showing no statistical difference in OsteoSense fluorescent intensity (Figure 7a,b). In some mice from both the CTRL and CKD groups, we observed an OsteoSense signal in the retina, which appeared more frequently in the CKD group (Figure 7c). However, the overall analysis of fluorescent signal intensity showed no difference between the two groups (Figure 7d). Additionally, CKD did not trigger calcification in the spleen as revealed by the OsteoSense staining and analysis (Figure 7e,f). Consistent with these findings, we observed no difference in mRNA levels of osteogenic markers in the brains and spleens of CTRL and CKD mice, indicating that CKD did not induce osteogenic transformation in these organs (Figure 7g,h).

## 3. Discussion

Here, we demonstrated that intravenous injection of OsteoSense EX750 imaging agent identifies organ-specific patterns of soft tissue calcification in an adenine-induced CKD model. We found that OsteoSense positivity is associated with increased mRNA expression of osteogenic transcription factors, suggesting the involvement of active osteogenic processes in soft tissue calcification. OsteoSense-based detection enables in vivo staining and the examination of calcification patterns in whole organs.

OsteoSense 750EX is a near-infrared fluorescent imaging agent specifically designed for the in vivo detection of hydroxyapatite, the primary mineral component of calcified tissues. OsteoSense dyes consist of a bisphosphonate group, which confers high affinity for hydroxyapatite, conjugated to a near-infrared fluorophore that enables optical imaging. Although the precise chemical structure of OsteoSense 750EX is proprietary, it is well-established that the bisphosphonate moiety ensures selective binding to sites of calcification, while the fluorophore component facilitates sensitive detection using near-infrared fluorescence imaging.

The OsteoSense imaging probe was initially designed to quantify bone remodeling and bone-associated diseases, including osteoporosis, arthritis, and bone metastases [10]. In connection with vascular calcification, Krohn et al. were the first to use OsteoSense to detect VSMC calcification in vitro [16]. Then, Wanga et al. detected calcification of the aortic root and the ascending aorta using OsteoSense staining on aortic sections derived from a mouse with Marfan syndrome [17]. First, Tóth et al. applied OsteoSense in vivo to detect aortic calcification in mice with adenine-induced CKD [19].

Patients with CKD exhibit a two- to fivefold increase in coronary artery calcification compared to healthy individuals of the same age. In CKD patients, vascular calcification serves as a powerful predictor of cardiovascular mortality [21]. Our study revealed that OsteoSense staining effectively identifies aortic calcification in CKD mice. Additionally, gross-tissue imaging using OsteoSense revealed distinct calcification patterns in the aorta, with the strongest fluorescence detected at the aortic arch and the abdominal region near the origin of the renal arteries.

In addition to the vasculature, calcification occurs in the kidneys in CKD patients. Research indicates that renal calcification starts early in the progression of CKD, with its prevalence increasing as the disease advances, exceeding 50% in patients with end-stage renal disease [22,23]. Additionally, studies showed that renal calcification contributes as a secondary pathogenic factor, accelerating the progression of CKD [22,23]. In this work, we showed profound calcification in the kidneys of CKD mice by OsteoSense staining. Kidney calcification was also confirmed with ARS and VKS methods.

Vascular calcification is an active process in which VSMCs gain osteoblast and chondrocyte-like gene expression profiles [23,24]. Here, we found that OsteoSense positivity is accompanied by increased mRNA expressions of osteo/chondrocyte transcription factors in the aorta and kidney, implicating the involvement of active osteo/chondrogenic trans-differentiation processes in the calcification in these tissues.

In addition to the aorta and the kidney, in this work, we investigated calcification in all other vital organs. Among them, we found CKD-associated calcification in the heart, lungs, and liver, whereas no calcification was detected in the organs derived from the CTRL mice. Calcification of the heart, especially the aortic and mitral valves, is frequently associated with CKD, with a prevalence between 25% and 59% among CKD patients on hemodialysis therapy [24]. Furthermore, valve calcification occurs 10–20 years earlier in CKD patients than in the general population, contributing to the high cardiovascular incidence and prevalence of cardiovascular events even in young CKD patients [25].

Pulmonary manifestations are common in CKD patients; among them, lung calcification was found in 3% of CKD patients using chest CT and X-ray, according to a recent study [26]. In this study, we found profound lung calcification in all the CKD mice, which agrees with a previous study by Westenfeld et al. in which the authors induced CKD by nephrectomy and high-phosphate diet and used VKS to detect calcifications in the kidney, heart, and lungs [27].

In humans, CKD-associated liver calcification is very rare. We found only seven previously reported cases in the literature. Based on these rare observations, diffuse liver calcification in CKD seems to be associated with liver hypoperfusion. Among the seven cases, liver calcification developed after a septic shock in two patients [28,29], and in two other patients, it developed on the clinical background of hemorrhagic shock [30]. The high sensitivity of the OsteoSense staining allowed us to detect diffuse liver calcification for the first time in a CKD mouse model. This method, therefore, provides a tool to study the currently unknown mechanism of this rare condition. We also detected elevations in the osteo/chondrogenic transcription factor mRNA levels in the calcifying organs, suggesting an actively regulated mechanism underneath the calcium deposition.

CKD impacts both the structure and function of the brain [31]. A recent study by Vinters et al. investigated the neuropathologic abnormalities in 40 autopsy brains originating from CKD patients [32]. Over half of the subjects experienced ischemic infarcts associated with large artery atherosclerosis and arteriolosclerosis [32]. Microvascular calcinosis was detected in more than a third of the subjects in the basal ganglia and/or endplate region of the hippocampus [32]. Bugnicourt et al. found a high prevalence of intracranial artery calcification in CKD patients [33]. In this work, we found no significant difference in brain calcification in CKD and control mice.

Regarding the eye, corneal and conjunctival calcifications are frequently seen in CKD patients undergoing hemodialysis [34,35]. Previous studies have reported that the prevalence of corneal and conjunctival calcification can be as high as 82.7–87.3% among CKD patients who have been on dialysis therapy for more than six months [35,36]. Retinal calcification has been described in several patients with a background of renal insufficiency [37,38,39]. Previously, using the adenine-induced CKD model in the calcification-prone DBA/2J mice, we observed calcification in the cornea [40]. Additionally, we provided evidence that high phosphate and calcium trigger osteo/chondrogenic differentiation of corneal epithelial cells in a Runx2-dependent manner [40]. In this work, we did not detect corneal calcification in C57BL/6 mice. Instead, we observed an OsteoSense signal from the retinal part of the eye, which was more frequent in the CKD group than in the control group, although the difference was not significant.

Splenic calcification is rare, but a recent multicenter study reported spleen calcification in dialysis patients diagnosed with calcific uremic arteriolopathy [41]. Here, we found that CKD did not trigger splenic calcification in C57BL/6 mice. In agreement with the OsteoSense staining, we found no increase in osteogenic marker mRNA levels in the brains and spleens of CKD mice.

Overall, this study demonstrated that OsteoSense staining is a sensitive and effective approach for detecting CKD-associated soft tissue calcification in C57BL/6 mice. The imaging agent reliably identified multiple organs undergoing calcification and revealed tissue-specific patterns of ectopic calcification. These findings are consistent with prior reports highlighting the systemic nature of calcification in CKD, particularly in vascular and renal tissues. Moreover, OsteoSense positivity was associated with increased levels of osteo/chondrogenic transdifferentiation markers. Interestingly, these osteo/chondrogenic markers were upregulated in a tissue-specific manner. For example, Msx2 and Bmp2 were the most upregulated in the calcifying aorta, but in the kidney, Runx2 and Sox9 were the most increased markers. During skeletal mineralization, bone formation proceeds through either endochondral ossification, which requires a pre-existing cartilage template, or intramembranous ossification, which occurs without a cartilage intermediate. Key osteogenic and chondrogenic transcription factors regulate these processes. Specifically, Sox9 and Runx2 are involved in endochondral bone formation, while Runx2 and Msx2 contribute to intramembranous ossification. Regarding vascular calcification, Runx2 is considered an essential transcription factor that promotes the osteogenic differentiation of VSMCs and interacts with other factors to drive the process. However, in general, the individual roles of these osteo/chondrogenic transcription factors in the process of soft tissue calcification are less understood and require further studies.

Importantly, OsteoSense also detected subtle calcification in organs such as the liver and heart, less frequently assessed in standard CKD models, underscoring its potential for uncovering previously unrecognized calcification sites. Given its specificity, non-destructive nature, and capacity for whole-organ assessment, OsteoSense is a powerful tool for in vivo and ex vivo evaluation of ectopic calcification. It holds promise for future preclinical studies aimed at understanding the mechanisms of soft tissue calcification and evaluating anti-calcification therapies. Ultimately, such approaches may inform new strategies to prevent or reverse ectopic calcification and improve cardiovascular and renal outcomes in patients with CKD.

## 4. Materials and Methods

### 4.1. CKD Induction in Mice

Animal experiments and procedures were conducted following institutional and national guidelines and were approved by the Institutional Ethics Committee of the University of Debrecen (registration number: 10/2021/DEMÁB). Mice were housed in cages with standard bedding and unlimited access to food and water under temperature- (22 °C) and light-controlled (12 h light/12 h dark) conditions. Male C57BL/6 mice (8–12 weeks old, n = 24) were randomly divided into two groups: Control (CTRL, n = 12) and CKD (n = 12). CKD was provoked by a two-phase diet as described previously [20]. In the first 6 weeks of the experiment, CKD mice received a diet containing 0.2% adenine and 0.7% phosphate, followed by 0.2% adenine and 1.8% phosphate for an additional 3 weeks (S8106-S075 and S8893-S006, respectively; Ssniff, Soest, Germany). Control mice received a standard chow diet.

### 4.2. OsteoSense Imaging and Quantification of Soft Tissue Calcification

We used OsteoSense 750EX dye (NEV10053EX, PerkinElmer, Waltham, Massachusetts, USA) to evaluate soft tissue calcification in mice. At the end of the experiment, we anesthetized the mice with isoflurane inhalation. Then, we injected them with OsteoSense dye in a dose of 80 nmol/kg body weight through the retro-orbital venous sinus. Imaging was performed 18 h post-injection. For this, we killed the mice with CO_2_ inhalation, took blood through a cardiac puncture, perfused the mice with 5 mL of ice-cold PBS, and collected all the organs. The organs were placed on a black sheet, and bright-field and fluorescent images were taken using an IVIS Spectrum In Vivo Imaging System (PerkinElmer, Waltham, MA, USA). OsteoSense 750EX has an excitation maximum at approximately 750 nm and an emission maximum near 770 nm, making it suitable for deep tissue imaging with minimal background signal due to low tissue autofluorescence in this spectral window. The absorption spectrum of the dye spans roughly 730–760 nm, with peak absorption centered near 750 nm. In our imaging protocol, excitation was performed at 750 nm, and fluorescence emission was collected at 770 nm in accordance with the manufacturer’s guidelines. Fluorescent signal intensity was measured and analyzed using the IVIS device’s built-in software.

### 4.3. Laboratory Analysis of Mice

Serum urea, creatinine, phosphate, and calcium levels were determined by kinetic assays on a Cobas^®^ c502 instrument (Roche Diagnostics, Mannheim, Germany).

### 4.4. Histology

The aortas and kidneys were fixed in 10% neutral-buffered formalin and embedded in paraffin. Subsequently, they were cut into 4 μm thick cross-sections. The sections were deparaffinized and rehydrated, followed by VKS and ARS using standard procedures. For VKS, the sections were placed in fresh 1% silver nitrate and exposed to UV light for 1 h, then washed with fresh 5% sodium thiosulfate for 5 min, and counterstained with neutral fast red for 5 min. For ARS, sections were stained for 5 min in 2% alizarin red S dye (pH 4.2). All sections were dehydrated using a graded alcohol series and mounted. All the sections were counterstained with Hematoxylin and Eosin (H&E).

### 4.5. Quantitative Real-Time PCR (qPCR)

Total RNA was isolated with TRI reagent, and cDNA was obtained using a High-Capacity cDNA Reverse Transcription Kit (Applied Biosystems, Waltham, MA, USA). qPCR was performed with a Bio-Rad CFX96 Real-time System (Bio-Rad, Hercules, CA, USA) using iTaqTM Universal SYBR ^®^ Green Supermix (Bio-Rad, Hercules, CA, USA) and the pre-designed primers listed in Table 1. The comparative Ct method was used to calculate the expression level of the transcripts, and hypoxanthine-guanine phosphoribosyl transferase (*Hprt*) was used for normalization as an internal control.

### 4.6. Statistical Analysis

Data are presented as mean ± SD with individual data points. Statistical analyses were performed with GraphPad Prism software (version 8.01). To calculate *p*-values, we used an unpaired one-tailed Mann–Whitney test. The value of *p* < 0.05 was considered significant.

## Figures and Tables

**Figure 1 ijms-26-08525-f001:**
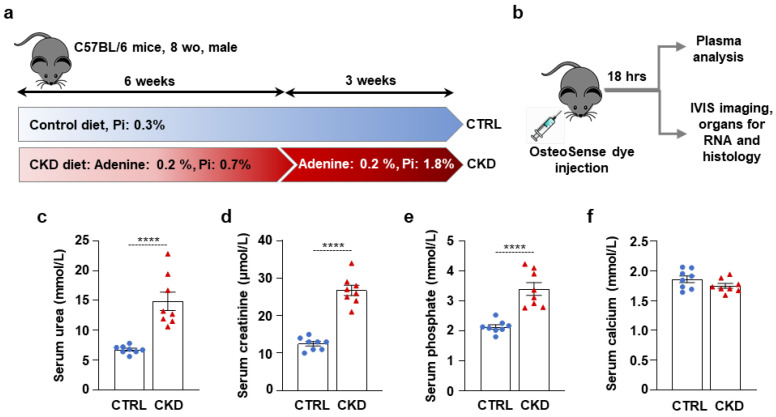
CKD induction in C57BL/6 mice fed with an adenine + high-Pi diet. (**a**,**b**) Scheme of the experimental protocol. (**c**) Plasma urea; (**d**) creatinine; (**e**) phosphate; and (**f**) calcium levels (n = 8). Data are expressed as mean ± SD. The Mann–Whitney test (unpaired, one-tailed) was used to calculate *p*-values. **** *p* < 0.001.

**Figure 2 ijms-26-08525-f002:**
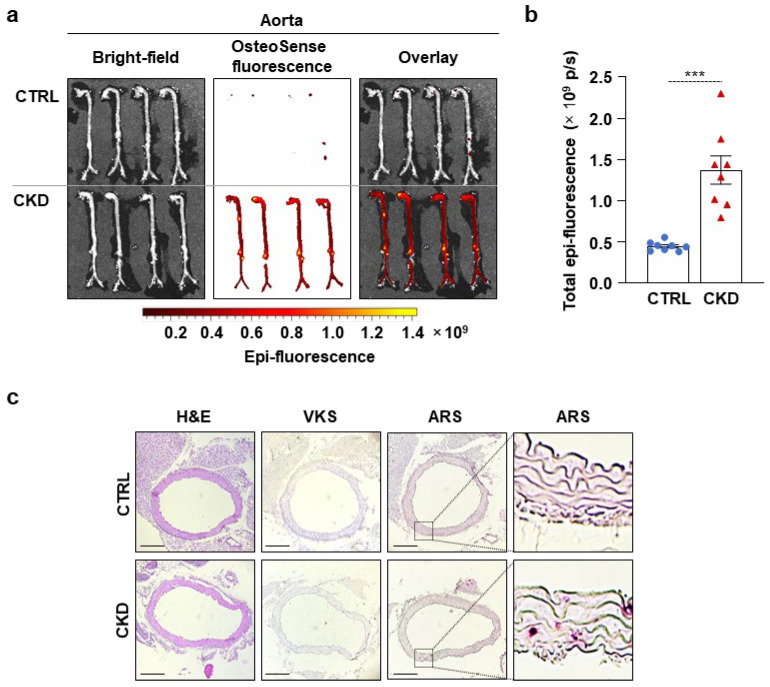
CKD is associated with calcification in the aorta in C57BL/6 mice. Mice were treated as shown in Figure 1a,b. (**a**) Bright-field and macroscopic fluorescence reflectance imaging of the aortas; (**b**) quantification of OsteoSense staining (n = 8). Data are expressed as mean ± SD. The Mann–Whitney test (unpaired, one-tailed) was used to calculate *p*-values. *** *p* < 0.005. (**c**) Histological analysis of the aorta obtained from CTRL and CKD mice (n = 4). Representative H&E, VKS, and ARS. Scale bars: 200 μm.

**Figure 3 ijms-26-08525-f003:**
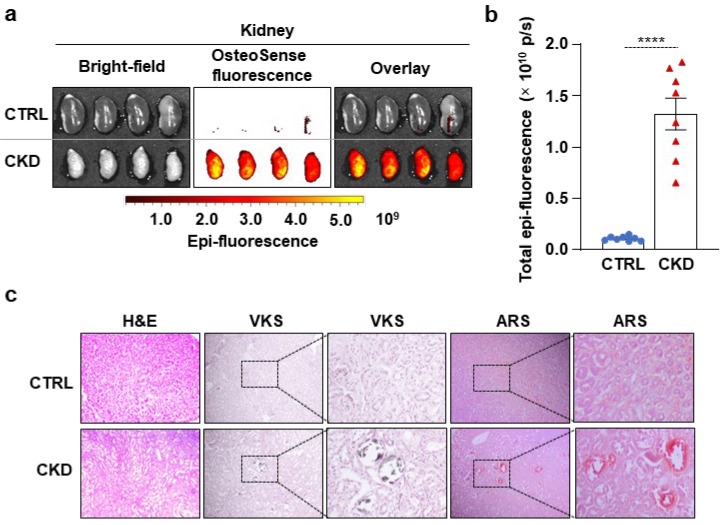
CKD is associated with calcification in the kidney in C57BL/6 mice. Mice were treated as shown in Figure 1a,b. (**a**) Bright-field and macroscopic fluorescence reflectance imaging of the kidneys; (**b**) quantification of OsteoSense staining (n = 8). Data are expressed as mean ± SD. The Mann–Whitney test (unpaired, one-tailed) was used to calculate *p*-values. **** *p* < 0.001. (**c**) Histological analysis of kidneys obtained from CTRL and CKD mice (n = 4). Representative H&E, VKS, and ARS. Scale bars: 200 μm.

**Figure 4 ijms-26-08525-f004:**
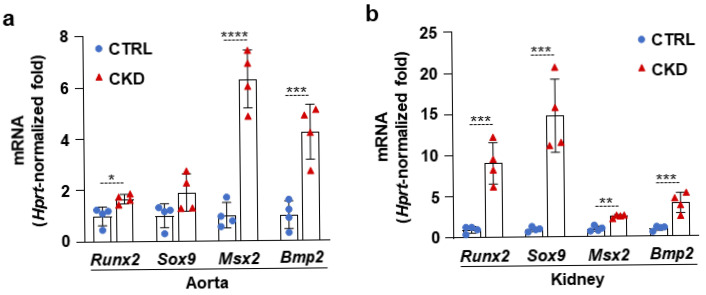
CKD is associated with the elevation of osteogenic marker mRNA levels in the aorta and kidney. (**a**,**b**) mRNA levels of osteogenic markers in the aorta and kidney in CTRL and CKD mice (n = 4). Data are expressed as mean ± SD. The Mann–Whitney test (unpaired, one-tailed) was used to calculate *p*-values. * *p* < 0.05, ** *p* < 0.01, *** *p* < 0.005, **** *p* < 0.001.

**Figure 5 ijms-26-08525-f005:**
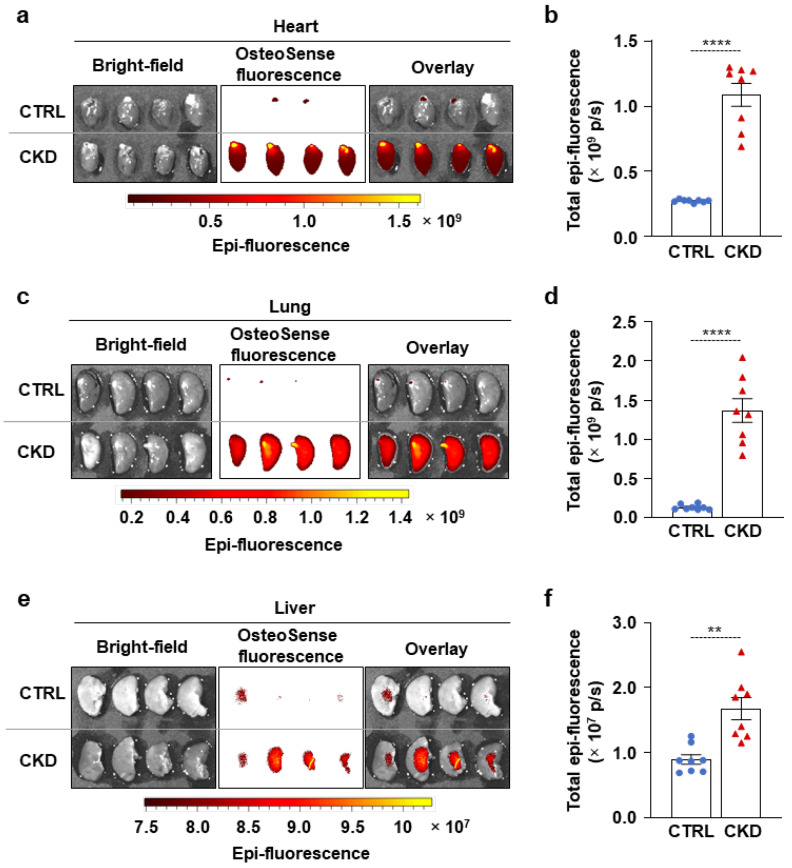
CKD is associated with calcification in the heart, lungs, and liver in C57BL/6 mice. Mice were treated as shown in Figure 1a,b. (**a**,**c**,**e**) Bright-field and macroscopic fluorescence reflectance imaging of hearts, lungs, and livers obtained from CTRL and CKD mice; (**b**,**d**,**f**) quantification of OsteoSense staining (n = 8). Data are expressed as mean ± SD. The Mann–Whitney test (unpaired, one-tailed) was used to calculate *p*-values. ** *p* < 0.01, **** *p* < 0.001.

**Figure 6 ijms-26-08525-f006:**
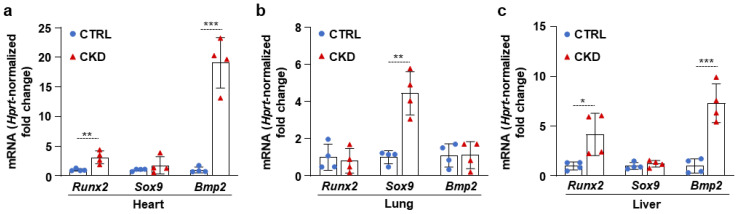
CKD is associated with the elevation of osteogenic marker mRNA levels in the heart, lung, and liver. (**a**–**c**) mRNA levels of osteogenic markers in the heart, lung, and liver in CTRL and CKD mice (n = 4). Data are expressed as mean ± SD. The Mann–Whitney test (unpaired, one-tailed) was used to calculate *p*-values. * *p* < 0.05, ** *p* < 0.01, *** *p* < 0.005.

**Figure 7 ijms-26-08525-f007:**
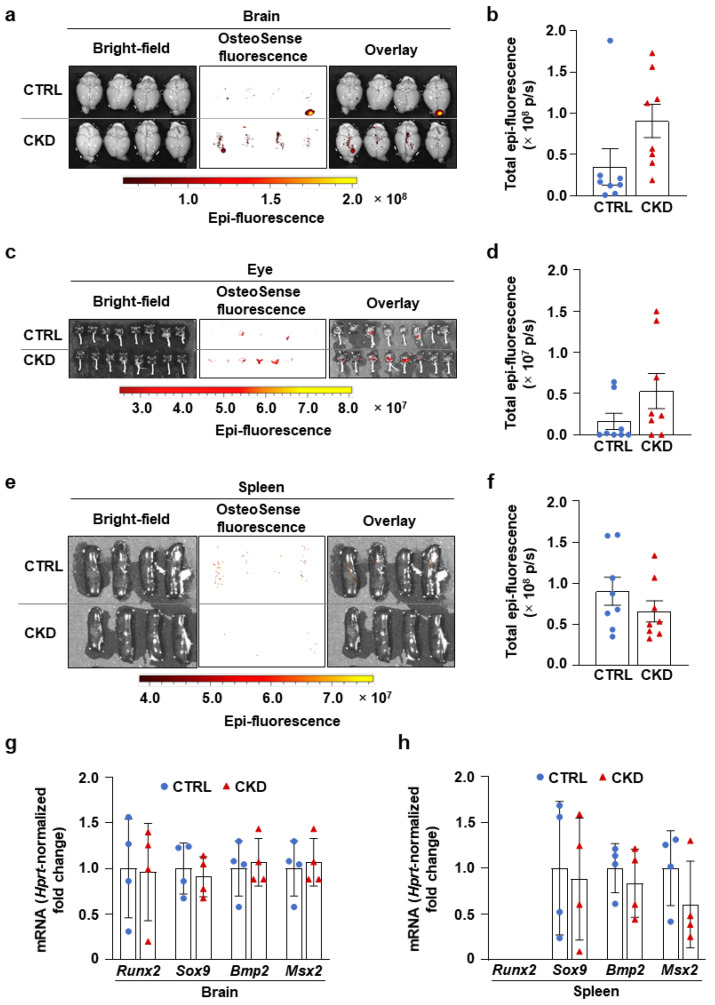
CKD does not trigger calcification in the brain, eye, and spleen in C57BL/6 mice. Mice were treated as shown in Figure 1a,b. (**a**,**c**,**e**) Bright-field and macroscopic fluorescence reflectance imaging of brains, eyes, and spleens obtained from CTRL and CKD mice; (**b**,**d**,**f**) quantification of OsteoSense staining (n = 8). (**g**,**h**) mRNA levels of osteogenic markers in the brain and spleen in CTRL and CKD mice (n = 4). Data are expressed as mean ± SD. The Mann–Whitney test (unpaired, one-tailed) was used to calculate *p*-values.

**Table 1 ijms-26-08525-t001:** List of primers.

Gene	Forward (5′-3′)	Reverse (5′-3′)
*Runx2*	GCATCCTATCAGTTCCCAATG	GAGGTGGTGGTGCATGGT
*Sox9*	GCTCTACTCCACCTTCACTTAC	TGTGTGTAGACTGGTTGTTCC
*Bmp2*	CGGACTGCGGTCTCCTAA	GGGGAAGCAGCAACACTAGA
*Hprt*	TCCTCCTCAGACCGCTTTT	CCTGGTTCATCATCGCTAATC
*Msx2*	AGGAGCCCGGCAGATACT	GTTTCCTCAGGGTGCAGGT

## Data Availability

The raw data supporting the conclusions of this article will be made available by the authors on request.

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
