# Peer review of "The OsteoSense Imaging Agent Identifies Organ-Specific Patterns of Soft Tissue Calcification in an Adenine-Induced Chronic Kidney Disease Mouse Model"

_ijms, 2025, doi:10.3390/ijms26178525_

Round 1
Reviewer 1 Report
Comments and Suggestions for Authors
Lente et al. presented their work utilizing a CKD mouse model to evaluate OsteoSense staining. Their results effectively demonstrate that OsteoSense is suitable for in vivo detection and quantification of calcification in soft tissues. Their calcification results were supported by gene expression analysis, showing that the expression of osteogenic genes was elevated in those tissues where calcification was detected by OsteoSense staining.
Notably, this is the first study to demonstrate CKD-related liver calcification in a mouse model.
Overall, the presented results convincingly show that OsteoSence can be used in vivo to detect soft tissue calcification in whole tissue, and is more sensitive than other staining methods.
Specific comments for the Authors:
- On Figure 2 and 3 the OsteoSense staining of whole tissues (aorta and kidney) and VKS and ARS staining of paraffin sections of the same tissues are shown. The OsteoSense staining proves the hypothesis of the authors that this method can be used to visualize and quantify calcification in whole tissues. However, to further establish the specificity and sensitivity of OsteoSense, it would be highly informative to include OsteoSense staining on paraffin sections of these same tissues, alongside the VKS and ARS staining. This would allow for a direct microscopic comparison of the different staining methods. Is it possible to detect OsteoSense fluorescence in the paraffin sections?
- Is there a reason why the expression of Msx2 was not measured in heart, lung and liver (Figure 6), only in aorta, kidney (figure 4) and brain and spleen (figure 7), and why osteogenic marker genes were not measured from the eye?
- The observed differences in osteogenic gene upregulation across the various calcifying tissues should be briefly discussed in the Discussion section. (Why are certain osteogenic genes elevated in some calcifying tissues but not others?)

Author Response
Comments 1: On Figure 2 and 3 the OsteoSense staining of whole tissues (aorta and kidney)
and VKS and ARS staining of paraffin sections of the same tissues are shown. The OsteoSense
staining proves the hypothesis of the authors that this method can be used to visualize and
quantify calcification in whole tissues. However, to further establish the specificity and
sensitivity of OsteoSense, it would be highly informative to include OsteoSense staining on
paraffin sections of these same tissues, alongside the VKS and ARS staining. This would allow
for a direct microscopic comparison of the different staining methods. Is it possible to detect
OsteoSense fluorescence in the paraffin sections?
Response 1: Thank you for your comment. We agree that demonstrating OsteoSense staining
in paraffin sections would provide valuable additional insight. However, our laboratory does
not currently have access to a fluorescence microscope capable of detecting OsteoSense, and
as such, we do not routinely perform this staining. In a collaboration, we conducted
OsteoSense staining on paraffin-embedded kidney sections. The staining was positive in
calcified kidneys but absent in control samples. Notably, the OsteoSense signal co-localized
with regions positive for Alizarin Red and von Kossa staining. These findings, however,
remain unpublished, and as the tissues were obtained from a separate experiment, we are
unable to include them in the present study.
Comments 2: Is there a reason why the expression of Msx2 was not measured in heart, lung
and liver (Figure 6), only in aorta, kidney (figure 4) and brain and spleen (figure 7), and why
osteogenic marker genes were not measured from the eye?
Response 2: Thank you for your question. We were unable to detect Msx2 expression in the
heart, lung, and liver tissues, as no amplification was observed. For this reason, we have not
included those results in the manuscript. The eye is a very heterogeneous tissue, with some
parts that are mostly acellular (lens). Also, the eye has a very low RNA content, which is why
we excluded the eye from the RNA analysis.
Comments 3: The observed differences in osteogenic gene upregulation across the various
calcifying tissues should be briefly discussed in the Discussion section. (Why are certain
osteogenic genes elevated in some calcifying tissues but not others?)
Response 3: Thank you for your comment. Based on the suggestion, we completed the
discussion with a paragraph as follows:
“Moreover, OsteoSense positivity was associated with increased levels of osteo/chondrogenic
transdifferentiation markers. Interestingly, these osteo/chondrogenic markers were
upregulated in a tissue-specific manner. For example, Msx2 and Bmp2 were the most
upregulated in the calcifying aorta, but in the kidney, Runx2 and Sox9 were the most increased
markers. During skeletal mineralization, bone formation proceeds through either
endochondral ossification, which requires a pre-existing cartilage template, or
intramembranous ossification, which occurs without a cartilage intermediate. Key osteogenic
and chondrogenic transcription factors regulate these processes. Specifically, Sox9 and Runx2
are involved in endochondral bone formation, while Runx2 and Msx2 contribute to
intramembranous ossification. Regarding vascular calcification, Runx2 is considered an
essential transcription factor that promotes the osteogenic differentiation of VSMCs and
interacts with other factors to drive the process. But in general, the individual roles of these
osteo/chondrogenic transcription factors in the process of soft tissue calcification are less
understood and require further studies.”

Reviewer 2 Report
Comments and Suggestions for Authors
This paper explored potential applications of contrast agent OsteoSense 750EX for soft tissue calcification detection in fluorescence imaging. This targeted application is new and original. Their results showed its effectiveness in the calcification detection in multiple types of vital organs, in particular in the aorta for early detection of cardiovascular diseases. This is significant. It may lead to a new efficient non-invasive imaging-based disease diagnosis tool development.
Some more information may need to be added to make the paper clearer. In the fluorescence imaging, what’s the excitation wavelength and what’s the fluorescence wavelength? Some more specific properties of the OsteoSense 750EX should be provided, such as its toxicity. The OsteoSense was intravenous inject in this study, it’s basically for the whole body imaging. Is it possible to target to a specific site in the body using this agent by doing some modification, for example, by adding a targeting molecular link in the agent to target to a specific site in the body?
Overall, this paper is written very well and research is solid. With some minor revision, it’s publishable
Author Response
Comments 1: Some more information may need to be added to make the paper clearer. In the fluorescence imaging, what’s the excitation wavelength and what’s the fluorescence wavelength? Some more specific properties of the OsteoSense 750EX should be provided, such as its toxicity. The OsteoSense was intravenous inject in this study, it’s basically for the whole body imaging. Is it possible to target to a specific site in the body using this agent by doing some modification, for example, by adding a targeting molecular link in the agent to target to a specific site in the body?
Response 1: Thank you for your comment. Based on the suggestion, we completed the method section with a paragraph as follows:
“OsteoSense 750EX has an excitation maximum at approximately 750 nm and an emission maximum near 770 nm, making it suitable for deep tissue imaging with minimal background signal due to low tissue autofluorescence in this spectral window. The absorption spectrum of the dye spans roughly 730–760 nm, with peak absorption centered near 750 nm. In our imaging protocol, excitation was performed at 750 nm, and fluorescence emission was collected at 770 nm in accordance with the manufacturer's guidelines.”
Additionally, we completed the discussion with the following paragraph:
“OsteoSense 750EX is a near-infrared fluorescent imaging agent specifically designed for the in vivo detection of hydroxyapatite, the primary mineral component of calcified tissues. OsteoSense dyes consist of a bisphosphonate group, which confers high affinity for hydroxyapatite, conjugated
to a near-infrared fluorophore that enables optical imaging. Although the precise chemical structure
of OsteoSense 750EX is proprietary, it is well-established that the bisphosphonate moiety ensures
selective binding to sites of calcification, while the fluorophore component facilitates sensitive
detection using near-infrared fluorescence imaging.”

Round 2
Reviewer 2 Report
Comments and Suggestions for Authors
The authors have provided enough information that readers may be interested. The paper is good enough for a publication
Author Response
Thank you very much for taking the time to review this manuscript.